# The Medial Surface of the Auricle: Historical and Recent Maps. What Are the Possible Expectations of the *“Thumb-Index Technique”*

**DOI:** 10.3390/medicines9020013

**Published:** 2022-02-17

**Authors:** Antonello Lovato, Francesco Ceccherelli, Giuseppe Gagliardi, Marco Postiglione

**Affiliations:** 1Pain Therapy Clinic, Via Sauro 29, 36045 Lonigo, Italy; 2A.I.R.A.S, Italian Association for the Research and the Scientific Update, 35100 Padua, Italy; info@ceccherelli.it (F.C.); gagliardi.beppe@gmail.com (G.G.); postiglia@yahoo.it (M.P.); 3Anesthesiology and Pain Therapy, Department of Anesthesiology and Reanimation, University of Padua, 35100 Padova, Italy; 4Postgraduate Specialization in Anesthesiology Specialist, Postgraduate Specialization in Sport Medicine Specialty, ULSS 5 Polesana, 45100 Rovigo, Italy; 5Postgraduate Specialization in Public Health and Preventive Medicine, Postgraduate Specialization in Physical and Rehabilitation Medicine, ULSS 5 Polesana, 45100 Rovigo, Italy

**Keywords:** auricular acupuncture, auriculotherapy, auriculomedicine, medial surface of the auricle, thumb-index technique

## Abstract

Introduction: The medial surface of the auricle (MSotA), as compared to the lateral, has been less studied and has limited consensus among schools of auricular acupuncture (AA) due to its small size, greater difficulty in carrying out an adequate physical examination on it, and less precise and limited agreement on its anatomical references as compared to the lateral surface. The thumb-index technique TIT is performed using a guiding finger (taking advantage of the anatomical conformation of the lateral surface) to guide the explorer finger (placed on the MSotA) to project the therapeutic areas and land marks on the MSotA. TIT could be considered useful and effective in AA to make the most of diagnostic and therapeutic MSotA potential. Methods: An investigation was carried out on the impact of TIT in AA practice through a survey collected from former AA students. Results: TIT showed a high consensus, and is used and appreciated by AA practitioners. Discussion/ Conclusions: To date, in AA, there is no thoroughly shared nomenclature for MSotA. TIT is simple and quick to project on to MSotA the well-coded lateral surface auricular maps from French or Chinese AA schools.

## 1. Introduction:

### 1.1. The Medial Surface of the Auricle (or MSotA)

The medial surface of the auricle (or MSotA), as compared to the lateral surface, has been less studied and has limited consensus among the different schools of auricular acupuncture (AA). This is due to the small size, the greater difficulty in carrying out an adequate physical examination on it, and the less precise and limited agreement on its anatomical references as compared to the lateral surface.

There has been much discussion on the innervation of the MSotA. From a study conducted on the innervation of the auricle of human cadavers, the medial surface can be divided into three parts [1]. The upper third of the MSotA is innervated by ATN (auriculo-temporal nerve of the trigeminal nerve) and GAN (great auricular nerve from the cervical plexus C2-3), the middle third is innervated mainly by ABVN (auricular branch of vagus nerve) and partially by GAN, and the lower third is innervated mainly by GAN and to some degree by ABVN, respectively.

In November 1990, WHO organized a working group on auricular nomenclature in Lyone (France). In this meeting, it was established that the MSotA can be divided into four parts as follows: posterior peripheral (PP), posterior intermediate (PI), posterior central (PC), and posterior lobular (PL), respectively [2]. To date, there is no fully shared nomenclature of the MSotA despite the efforts made by the Word Federation of Chinese Medicine Societies (WFCMS) during the last meeting held in Lyon in June 2012 [3].

### 1.2. Historical Maps

In China, great importance has always been given to AA diagnosis based on the inspection of the auricle. AA in China has never been systematically defined with the use of maps and has never had its own development (unlike somatic acupuncture), although it was performed earlier than in France. The auricle was mentioned in ancient Chinese medicine texts, such as the *Yin–Yang Eleven Channel Moxibustion* (*Yinyang Shiyi Mai Jiujing*), written during the Warring States period and Spring and Autumn Period, which is between 770 and 221 BC [4]. A first description, though very primitive, of the auricle appears in the *Linh Shu* (474-221 BC).

In a book written during the Ming Dynasty (1368AD-1644AD) by Zhou Yufan called *Massage Technique for Children* (*Xiaoer Anmo Shu*), for the first time, the five Zang organs (i.e., heart, liver, spleen, lung, and kidney) theory was described as regards to their correspondence to the MSotA [4]. Based on this theory, Zhenjun Zang produced the first AA map in a book published in 1888 *Essential Techniques for Massage* (*Lizheng Anmo Yaosu*) [5].

### 1.3. Recent Maps

Starting from clinical observations, in 1957, P. Nogier proposed the somatotopy of the auricle, creating a map that has become a reference for all practitioners from East to West. The map was drawn by G. Bachmann, based on the proposal of P. Nogier, and published in a German magazine *Deutsche Zeitschrift für Akupunktur* [6,7]. This map was used as the gold standard for AA, although there is no mention of the MSotA as shown in Figure 1 and Figure 2.

This significant idea of P. Nogier traveled the world and arrived in China, where, based on Nogier’s work, a very simple first map was drawn, which was probably reserved for military doctors. Chinese doctors learned to use AA as a “microsystem” and, as time passed and experience increased, more auricular points were discovered.

In 1971, H. Jerricot, an illustrious French physician, cited the Chinese map of the auricle, which still only described the lateral surface [8], as shown in Figure 3.

In 1975, given the lack of information on the MSotA in the first 1957 auricle map, Nogier, together with Boudiol and Bahr, published a map representing the musculoskeletal system and visceral areas on the MSotA [9].

In 1981, Marco Romoli proposed and published his first septogram. In this first septogram, Romoli also drew the medial surface. Strictly speaking, the septogram cannot be defined as a map but is rather a tool used to delineate the skin changes detected at the auricle on a two-dimensional image to formulate an accurate AA diagnosis [10].

In 1988, the Chinese Association of Acupuncture and Moxibustion (CAAM) published an article called “The project of the standardization of auricular acupuncture points”, in which standardization of the lateral and medial surface of the auricle was proposed. The lateral surface was divided into areas that largely reflected the anatomy of the lateral surface of the auricle, integrated by AA points placed in areas that were not visible on the map. The medial surface, on the other hand, recalled the Chinese historical map, which was connected with the five Zang organs. This map is still recognized as the Nanjing Map and represents a reference point for Chinese standardization [11].

In 1990, Terry Oleson proposed his second map, which was considered innovative in many ways. It highlighted the hidden parts of the auricle on both the lateral and medial surfaces. Furthermore, the mapping of auricle areas by Oleson was characterized by being very adaptable to any auricle because it had no fixed areas, instead showing these areas as being proportionally divided into the individual parts of the auricle similar to the “CUN” measurement used in somatic acupuncture. Oleson’s map (as Romoli’s septogram) cannot be properly defined as a map as the author did not propose his own points but delineated the position of points from the French and Chinese schools. The map essentially considered (with one exception on the lateral surface and three on the medial surface) the nomenclature of the areas of the auricle established by WHO in 1990 in Lyon (France) [12]. Oleson recognized and described 24 auricular landmarks (LMs) on the lateral surface, which are identified as well-defined areas used as landmarks for researching specific therapeutic auricle zones. LMs are not necessarily therapeutic zones and, in a study carried out by Oleson himself, they were found to be reliable reference points since the distances between the various LMs were found to be fairly constant with a relatively low standard deviation, regardless of race, age, and sex [13]. This map is shown in Figure 4.

In 2003, Marco Romoli published his book on AA *Agopuntura Auricolare* (published by UTET in 2003), in which the septogram was modified and the image of the MSotA was dropped [14]. Furthermore, it became “flexible” since the rays of the septogram were not fixed but created by drawing three lines that connected specific anatomical references to the point zero, thus dividing the auricle into three macro-sectors. Then, each macro-sector was divided into a specific number of smaller sectors by lines drawn from the point zero, which were proportional to the three macro-sectors previously delineated, as shown in Figure 5.

In 2010, David Alimi presented a new nomenclature and AA cartography to WHO through the World Federation of Chinese Medicine Societies (WFCMS). This AA map was centered on the prime zero point (or corpus callosum point) rather than on the point zero, unlike Romoli’s septogram. This map marked the points on the lateral surface in a different way according to their localization and a different color was used based on the embryological origin of the tissue represented therein. On this map, the AA points on the MSotA were all drawn with the same symbol (i.e., a square) to mark the difference with points found on the lateral surface of the auricle. The coloring of AA points represented the embryological origin of the tissue represented therein. This map consisted of 20 rays all with the same angle. The limit was that this map was not as “flexible” as others since it had to be customized to each patient and it did not highlight hidden areas as well as Oleson’s map [15,16]. This map is shown in Figure 6.

A recent publication by Lei Wang et al. aimed to verify the correspondence between AA points as proposed by the Chinese system (as represented by the World Federation of Acupuncture-Moxibustion Societies or WFAS) and as proposed by the European system (i.e., according to Nogier and Bahr) [17]. The authors concluded that 24 AA points have the same name and position on the auricle. Furthermore, 25 areas or points maintain the same nomenclature but represent body parts that are different between the 2 systems and therefore have different therapeutic indications.

In 2016, Pei-Jing Rong analyzed 9 methods to localize AA points, with the aim of highlighting the strengths and disadvantages of each of these methods [18]. The paper concluded that the international method proposed by the WFAS based on lines and points was an appropriate method. The same paper recognized that the areas described and proposed by the WFAS for the MSotA were too large and that an effort to improve the localization of the AA points of the MSotA was desirable.

Andres Wirz-Ridolfi explained that for AA to be effective, it must be very precise [4]. This concept has been underlined by works carried out with functional magnetic resonance imaging (fRMN) in recent studies, such as those by David Alimi in 2002 and 2014 [19,20] and by Marco Romoli in 2015 [21]. The author stressed the need to have better and more comparable maps to obtain greater international consensus on AA points and greater international credibility. Furthermore, the author stated that scientific recognition is certainly important but the final goal (to be considered even more important) is to obtain good therapeutic results, which is possible only if the correct AA points of the auricle are treated (i.e., punctured by the acupuncture needles or irradiated by LASER).

### 1.4. Critical Points to Mapping the MSotA

The MSotA could be described as an undefined area as it is the least studied of the auricle. The anatomical structure, small size (as compared to the lateral surface), and fewer anatomical references contribute to making MSotA be the least studied and known area of the auricle. The map Oleson published in 1990 helped to highlight the hidden areas of the medial surface. Oleson, in his most recent map (Figure 7), divided the medial surface into five parts: posterior periphery (PP), posterior concha (PC), posterior triangle (PT), posterior groove (PG), and posterior lobe (PL), respectively. Therefore, there is no shared nomenclature of the MSotA to date [12].

Another limit of Oleson’s map is that it shows the MSotA only as “open” and therefore it does not correlate with the real image of the patient. Similarly to the lateral surface, it would be desirable to have a natural and “open” image.

If the concept of a three-dimensional representation of the organs and anatomical structures on the auricle is achieved, considering the lateral and medial surfaces with completely different maps appears difficult. On the other hand, drawing two maps, one for the lateral surface and one for the MSotA, means considering that the two surfaces of the auricle are completely disconnected from each other. Furthermore, it means that the therapeutic area has a superficial and two-dimensional value (Figure 8).

### 1.5. The Thumb-Index Technique

To overcome these difficulties and to make the most of the diagnostic and therapeutic potential of MSotA, the authors propose a method that can be used in everyday AA practice. The method aims to project the points and areas of the auricle and, where possible, the LMs (auricular landmarks) described by Oleson of the lateral surface on the MSotA. The method proposed is the thumb-index technique (or TIT).

This method uses a guiding finger and an explorer finger. The guiding finger takes advantage of the anatomical conformation of the lateral surface to guide the explorer finger placed on the MSotA. This allows the practitioner to project the therapeutic areas and the LMs on the MSotA.

The result is a narrower or flattened projection of the homunculus on the MSotA while the practitioner can never lose the overall understanding of the area explored and, above all, avoids two distinct and discordant results when examining the lateral surface and the MSotA.

This simple and quick technique also allows the points of the lateral surface identified by both the French and Chinese Schools of AA to be projected on the MSotA. The TIT is shown in Figure 9.

The purpose of this article is to understand the new, quick, and simple thumb-index technique and whether it could be considered useful and effective to make the most of the diagnostic and therapeutic potential of the medial surface of the auricle in auricular acupuncture.

## 2. Materials and Method

An investigation was carried out on the impact of TIT in AA practice. The investigation was carried out by administering a set of questions on TIT to medical doctors who practice AA. The subjects selected were medical doctors, who were former AA course students of the past 5 years (from 2016 to 2020) who obtained the AA practitioner diploma from the AIRAS AA School in Padova (Italy).

A request to participate in the study was sent via email together with the set of 7 questions to answer.

The set of 7 questions were formulated as follows:(1)How many years have you been practicing auricular acupuncture (AA)?(2)Do you use the MSotA when practicing AA?(3)On a scale from 1 to 10 (where 1 is easy and 10 is very difficult), how difficult is it for you to locate therapeutic areas on the MSotA?(4)On a scale from 1 to 10 (where 1 is not important and 10 is very important), how important would it be to have a quick and simple method to be able to locate therapeutic areas on the MSotA?(5)Do you know the thumb-index technique (or TIT)? After a brief explanation of the TIT, the last two questions were asked.(6)On a scale from 1 to 10 (where 1 is not much and 10 is a lot), how much do you think TIT would help locate therapeutic areas on the MSotA?(7)On a scale from 1 to 10 (where 1 is never and 10 is always), now that you know this method would you include it in your everyday AA practice?

The answers were recorded and evaluated. The gender and age of the subjects who answered the survey were recorded.

## 3. Results

A total of 108 subjects were contacted and asked to answer the set of questions. A total of 70 subjects completed the survey, of which 58.6% were male and 41.4% were female with a mean age 51.94 years (SD ± 12.07, maximum and minimum age of 70 and 32 years, respectively).

Table 1 includes the results of the survey sent to the investigated subjects. The results listed include the mean score, standard deviation (SD), mode, and maximum and minimum score. Questions are listed in the materials and methods by number.

The subjects declared had been practicing auricular acupuncture for a mean of 4.81 years (SD 4.92, maximum and minimum score of 20 and 1, respectively). Of the subjects interviewed, 80% admitted to using the MSotA in their everyday AA practice. More than 80% of subjects agreed on the importance of having a quick and simple method to be able to locate therapeutic areas on the MSotA. As many as 81% stated that they had knowledge of TIT.

Nearly all subjects agreed, when answering the last question of the survey, that they would include TIT in their everyday AA practice now that they knew the method, with a mean score of 8.25 (SD 1.17, maximum and minimum score of 10 and 5, respectively) on a scale from 1 to 10 (where 1 is never and 10 is always).

## 4. Discussion and Conclusions

To date, there has been no nomenclature of the MSotA areas that is thoroughly shared by the various AA schools. Oleson’s map has the advantage of being flexible as it adapts to the auricle regardless of its shape and size. Oleson described 24 LMs on the lateral surface, which are reference points used for searching for further therapeutic or diagnostic areas. However, LMs are not described for the MSotA; thus they must eventually be deduced by the AA practitioner.

The acupuncture point has a three-dimensional value, being primarily located at sites that correspond to the neurovascular bundles of blood plexuses and myelinated and unmyelinated nervous fibers. It is reasonable to believe that drawing the acupuncture point on a map of the lateral surface of the auricle and, again, on a map on the MSotA downgrades its value from its three-dimensional concept to a two-dimensional interpretation.

To circumvent these critical points and avoid tedious and time-consuming practices, the practitioner can easily find the corresponding reference points of the lateral surface of the auricle and project them on the MSotA using the TIT. TIT is considered to be a well-accepted method since it is a simple and quick way to project the well-coded auricular maps of the lateral surface on the MSotA, regardless of whether they are from French or Chinese AA schools. Furthermore, this method also considers the three-dimensional structure of the acupuncture point and the likely three-dimensional representation of organs and systems on the auricle surface.

It is reasonable to believe that the mapping the MSotA for further somatotopies is a difficult and inefficient exercise that can easily be overcome with the thumb-index technique.

## Figures and Tables

**Figure 1 medicines-09-00013-f001:**
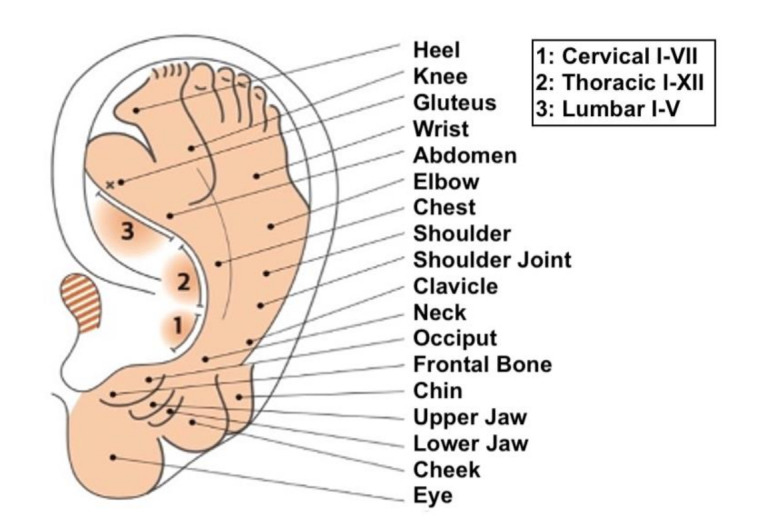
The historical map of P. Nogier drawn by Bachman. Image taken from “Agopuntura Auricolare teoria e clinica”, Antonello Lovato, Noi Edizioni, 2019.

**Figure 2 medicines-09-00013-f002:**
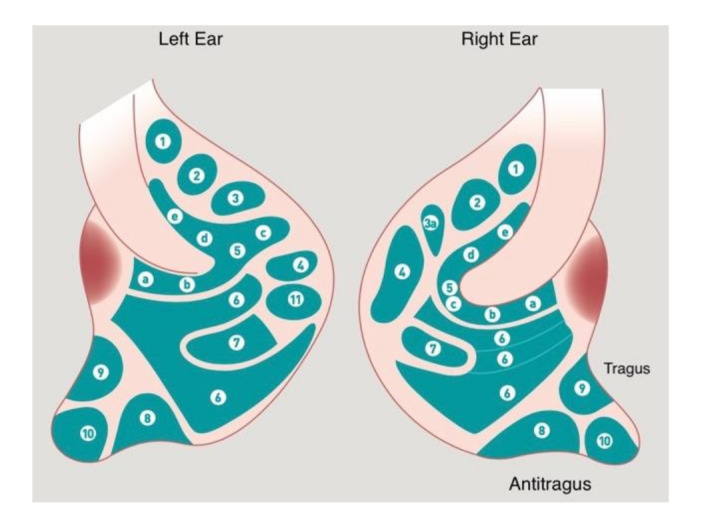
The historical map of P. Nogier drawn by Bachman. Legend of Figure 2: 1. Bladder; 2. Kidney; 3. Pancreas; 3a. Gallbladder; 4. Liver; 5a. Esophagus; 5b. Cardia; 5c. Stomach; 5d. Small Intestine; 5e. Large Intestine; 6. Lung; 7. Heart; 8. Subcortex; 9. Internal Nose; 10. Endocrine; 11. Spleen.

**Figure 3 medicines-09-00013-f003:**
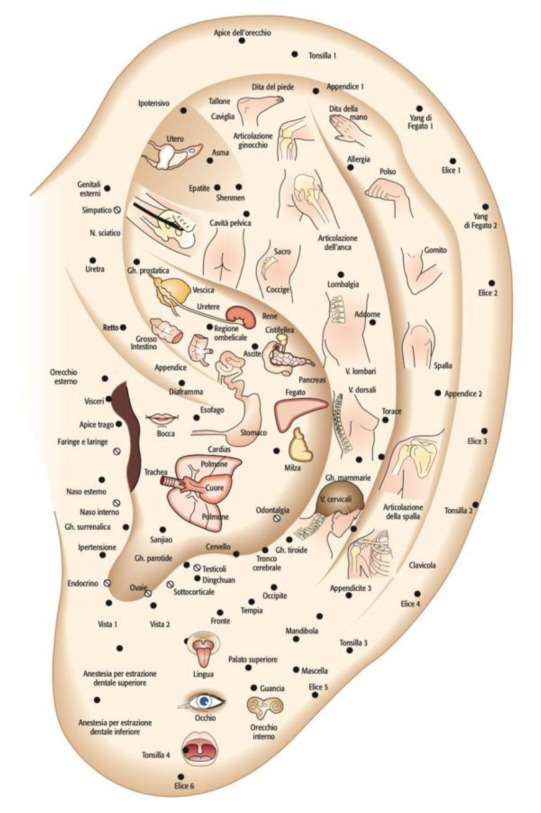
Chinese map cited by MD Henri Jarricot, 1971.

**Figure 4 medicines-09-00013-f004:**
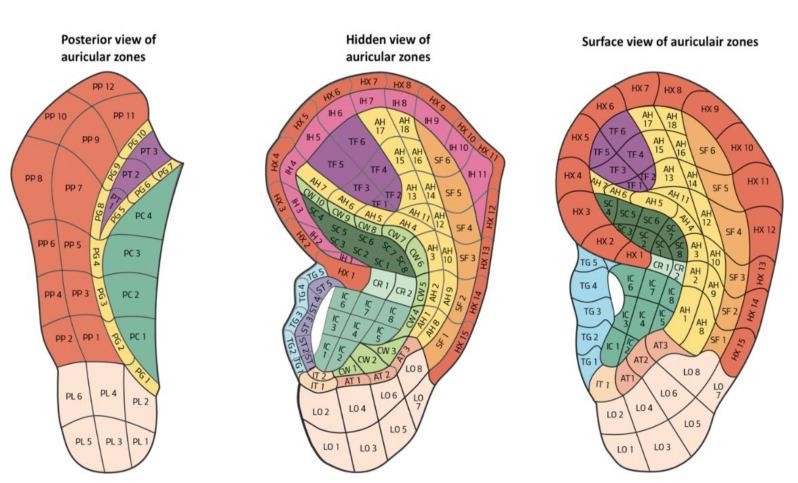
Oleson’s map highlighting the hidden parts of the auricle, 1980.

**Figure 5 medicines-09-00013-f005:**
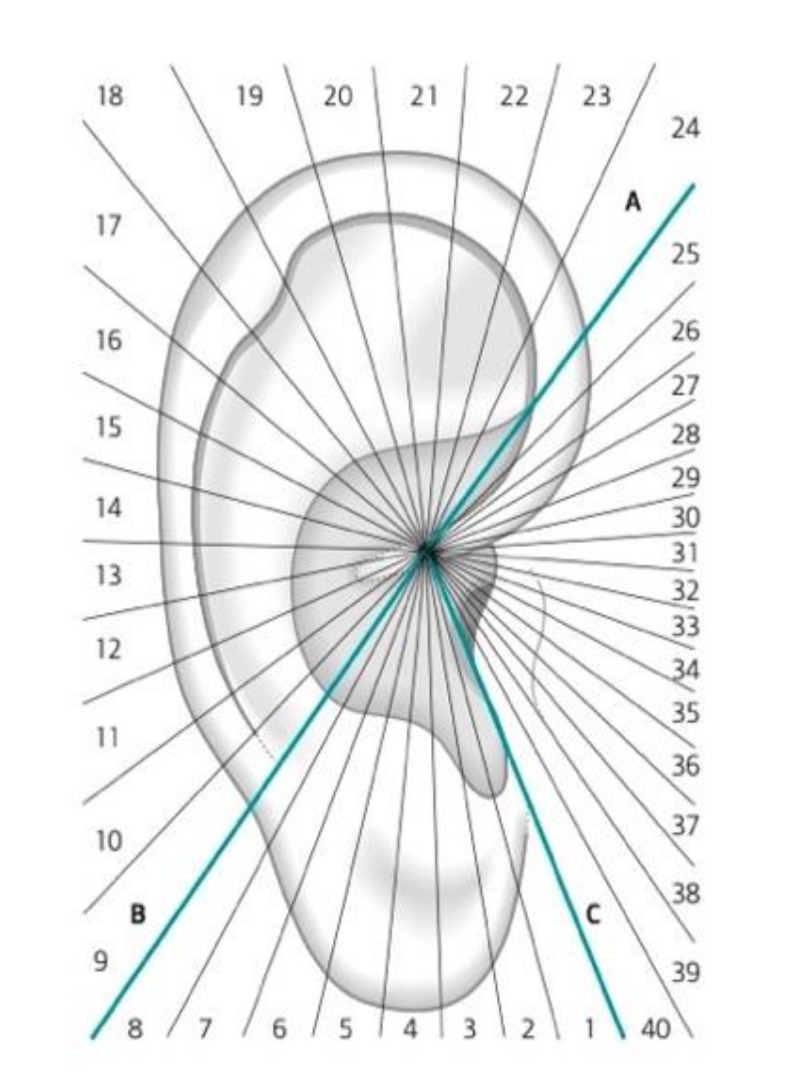
Second version of Romoli’s septogram, 2003.

**Figure 6 medicines-09-00013-f006:**
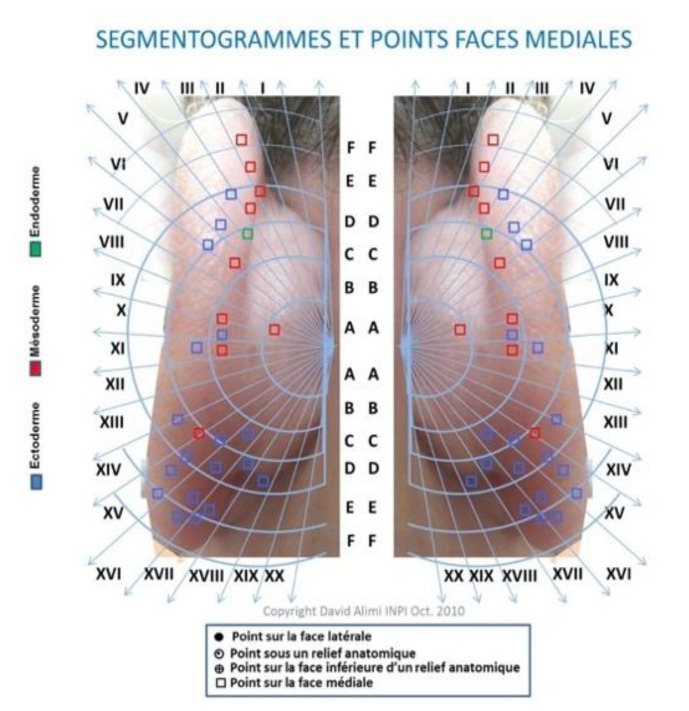
The medial surface of the map by D. Alimì, 2010.

**Figure 7 medicines-09-00013-f007:**
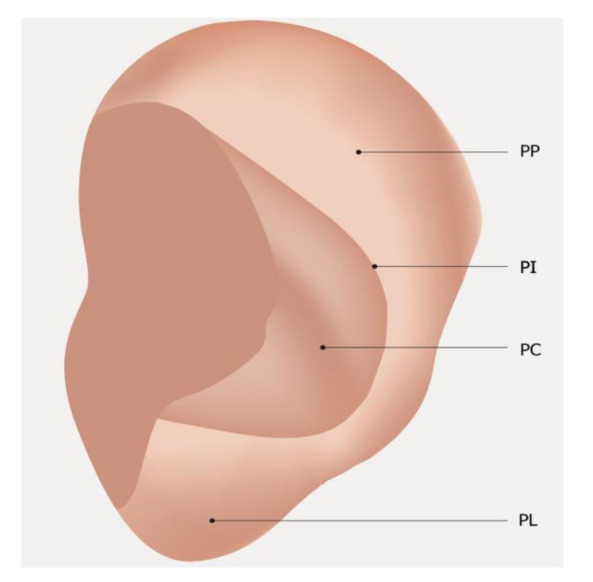
Medial surface nomenclature WHO Lyon 1990. Legend: PP: Posterior Peripheral; PI: Posterior Intermediate; PC: Posterior Central; PL: Posterior Lobular.

**Figure 8 medicines-09-00013-f008:**
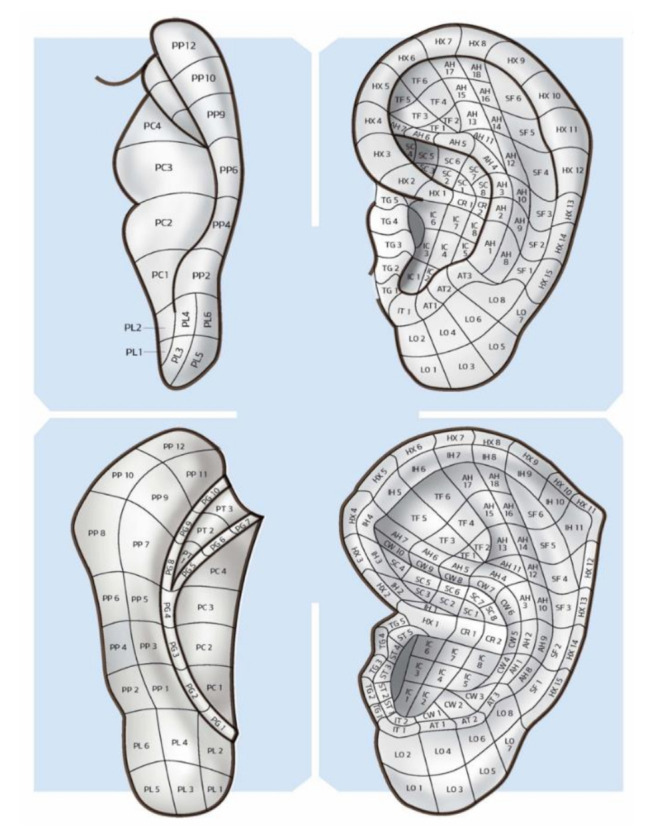
Lateral and medial surface of the auricle by A. Lovato 2019, with the division into areas according to the nomenclature of T. Oleson.

**Figure 9 medicines-09-00013-f009:**
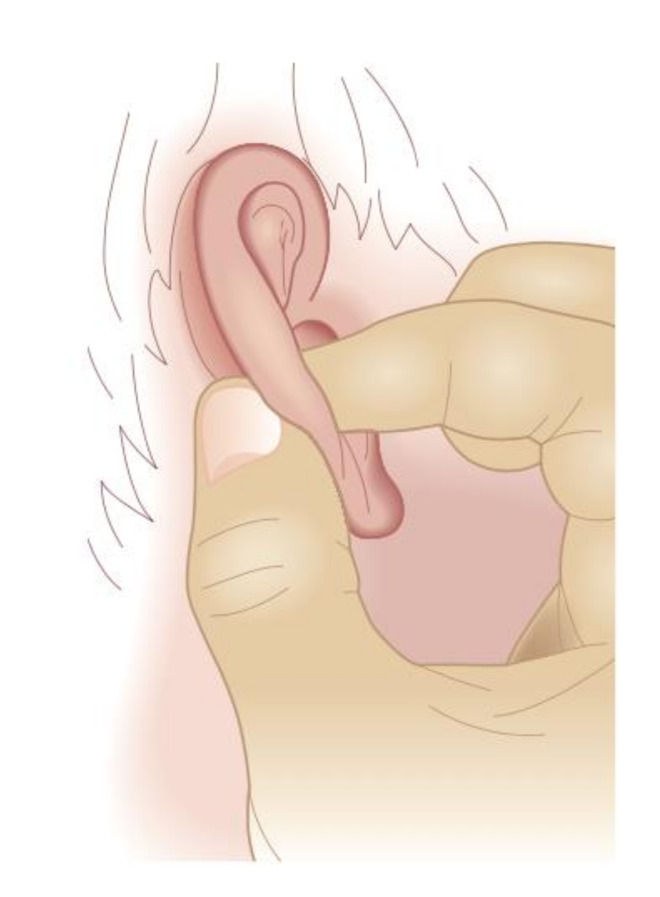
The thumb-index technique (TIT).

**Table 1 medicines-09-00013-t001:** The results obtained from the survey are illustrated.

Question	Mean	SD	Max Score	Min Score
1	4.81	±4.92	20	1
3	6.14	±2.17	10	1
4	8.25	±2.25	10	1
5	0.81	±0.39	1	0
6	8.17	±1.85	10	1
7	8.25	±1.17	10	1

## Data Availability

Not applicable.

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
