# Peer review of "The Medial Surface of the Auricle: Historical and Recent Maps. What Are the Possible Expectations of the “Thumb-Index Technique”"

_medicines, 2022, doi:10.3390/medicines9020013_

Round 1

Reviewer 1 Report

The authors gave a clear and great review on the history of Auricular Acupuncture and proposed that Thumb-Index Technique could be used in Auricular Acupuncture because it is simple and quick to project on MSotA the well-coded lateral 24 surface auricular maps. However, I did not see any experiments or data to support this opinion. Since this is the novelty of this paper, I would like authors could provide more related evidence to support it.  

Author Response

This submission follows a previous one done on 2011.11.13. Previous submission was reviewed. Points were made by reviewers as stated on the  2 Review Report Forms. These points have been thoroughly addressed through a meticulous redrafting of the paper by the authors. Since TIT is a tool to be used in everyday practice of AA to take advantage of the MSofA, to assess this impact of the method described an appropriate investigation was carried out. The investigation has been done asking a set of questions on TIT to colleagues who practice AA.  The group investigated included former students of the AA course organized by AIRAS in the last 5 years. Data obtained have been analyzed to produce results worth mentioning. The authors hope redrafting the paper has achieved the characteristics suggested on the review report forms (i.e. appropriate research design, adequately described methods, clearly presented results and conclusions supported by the results) and fulfilled reviewers comments and suggestions. MD Antonello Lovato

Reviewer 2 Report

It would be more scientific with implementation of control group or any data regarding therapeutic benefit of presented method.

Author Response

This submission follows a previous one done on 2021.13.11 Previous submission was reviewed. Points were made by reviewers as stated on the  2 Review Report Forms. These points have been thoroughly addressed through a meticulous redrafting of the paper by the authors. Since TIT is a tool to be used in everyday practice of AA to take advantage of the MSofA, to assess this impact of the method described an appropriate investigation was carried out. The investigation has been done asking a set of questions on TIT to colleagues who practice AA.  The group investigated included former students of the AA course organized by AIRAS in the last 5 years. Data obtained have been analyzed to produce results worth mentioning. The authors hope redrafting the paper has achieved the characteristics suggested on the review report forms (i.e. appropriate research design, adequately described methods, clearly presented results and conclusions supported by the results) and fulfilled reviewers comments and suggestions.  MD Antonello Lovato

Round 2

Reviewer 1 Report

The quality of manuscript has been improved after revision. 

Reviewer 2 Report

Materials, methods, results, discussion and conclusion are very short at all....